# PYGM mRNA expression in McArdle disease: Demographic, clinical, morphological and genetic features

Alzira A. S. Carvalho[1]*, Denise M. Christofolini[2,3], Matheus M. Perez[4], Beatriz C. A. Alves[4], Itatiana Rodart[2], Francisco W. S. Figueiredo[5], Karine C. Turke[6], David Feder[7], Marcondes C. F. Junior[8], Ana M. Nucci[8], Fernando L. A. Fonseca[4]

1 Departamento de Neurociências- Laboratório de doenças neuromusculares da Centro Universitário Saúde ABC, Santo Andre, São Paulo, Brazil, 2 Instituto Ideia Fértil de Saúde Reprodutiva do Centro Universitário Saúde ABC, Santo Andre, São Paulo, Brazil, 3 Departamento de genética do Centro Universitário Saúde ABC, Santo Andre, São Paulo, Brazil, 4 Laboratório de análises clínicas do Centro Universitário Saúde ABC, Santo Andre, São Paulo, Brazil, 5 Departamento de estatística do Centro Universitário Saúde ABC, Santo Andre, São Paulo, Brazil, 6 Medical student of Centro Universitário Saúde ABC, Santo Andre, São Paulo, Brazil, 7 Departamento de farmacologia da Centro Universitário Saúde ABC, Santo Andre, São Paulo, Brazil, 8 Departamento de Neurologia da UNICAMP, Campinas, SP, Brazil

* alzirasiqueiracarvalho@gmail.com

## Abstract

### Introduction

McArdle disease presents clinical and genetic heterogeneity. There is no obvious association between genotype and phenotype. *PYGM (*muscle glycogen phosphorylase gene) mRNA expression and its association with clinical, morphological, and genetic aspects of the disease as a set have not been studied previously.

### Methods

We investigated genetic variation in *PYGM* considering the number of PTCs (premature termination codon) per sample and compared mRNA expression in skeletal muscle samples from 15 patients with McArdle disease and 16 controls to PTCs number and different aspects of the disease.

### Results

The main variant found was c.148C>T (PTC—premature termination codon). Patients with two PTCs showed 42% mRNA expression compared to the control group. Most cases showed an inversely proportional relation among PTCs and mRNA expression. Association between mRNA expression and other aspects of the disease showed no statistically significant difference (p> 0.05).

### Discussion

mRNA expression is not useful as a predictor factor for the prognosis and severity of the disease. Different mechanisms as post-transcriptional events, epigenetics factors or protein function may be involved.

**Data Availability Statement:** All relevant data are within the manuscript and its Supporting Information files.

**Funding:** The authors received no specific funding for this work.

**Competing interests:** NO: The authors have declared that no competing interests exist.

**Abbreviations:** CK, creatine kinase; CNV, copy number variation; ExAC, Exome Aggregation Consortium; H&E, hematoxiline and eosine; HGMD®, Human Gene Mutation Database; PTC, premature termination codon; PYGM, glycogen phosphorylase gene; ROC, receiver operating characteristic; RT-qPCR, real time quantitative polymerase chain reaction; RPL13α, *ribosomal protein L13a*.

## Introduction

McArdle disease is the most common disorder of muscle metabolism, although it is a rare disease with a prevalence of one in 100,000 individuals.[1–3] The pathology is restricted to skeletal muscle, and so the symptoms include myalgia, fatigue, muscle contracture, exercise intolerance, rhabdomyolysis, myoglobinuria, and in more severe cases, acute renal failure. Additionally, an elevated creatine kinase (CK) level is always present in those patients at rest and exercise [3]. Additionally, the sedentariness may aggravate the exercise intolerance by further reducing the limited oxidative capacity caused by blocked glycogenolysis. A complete absence of myophosphorylase activity and presence of subsarcolemmal glycogen storage vacuoles in muscle fibers and/or identification of causative variants in the *PYGM* (glycogen phosphorylase gene—OMIM #232600) are used to confirm clinical diagnosis [3–5].

The condition is an autosomal recessive disorder caused by pathogenic sequence variants in *PYGM* associated to significant clinical and genetic heterogeneity among patients [3,6,7]. However, there is no obvious association between genotype and phenotype [6–8]. Thus, the factors that determine the severity of this disorder remain to be clarified.

With regard to *PYGM* mRNA expression and clinical, morphological, and genetic aspects of the disease is a little-known fact. Based on this knowledge gap, we hypothesize that it could determine different features of the disease. In order to test our hypothesis, we evaluated the association of *PYGM* mRNA expression in patients with phenotypic variations of McArdle disease.

## Methods

### Study design

This study was cross-sectional and prospective. Clinical and laboratorial data, as well as biopsy data were obtained from medical records. mRNA expression and genetic analyses were newly acquired for the study.

### Setting

15 patients previously diagnosed with McArdle disease were selected between 2015 and 2017 from the outpatient neuromuscular clinic of Centro Universitário Saúde ABC, in the Southeast region of Brazil.

One control group, composed by 16 patients with some clinical neuromuscular complaints and muscle biopsy with nonspecific findings, was used to compare results of mRNA expression.

The diagnostic inclusion criteria of McArdle disease included typical clinical history of exercise intolerance with persistent elevated CK at rest and absence of myophosphorylase staining on muscle biopsy [5]. In addition, a positive result for MIFET (Modified Ischemic Forearm Exercise Test), characterized by flat lactate levels and an exaggerated rise of ammonia during exercise, [9] was also considered to the diagnosis. All patients showed positive PAS staining inside the subsarcolemmal vacuoles. Patients met all inclusion criteria to be part of the study.

### Demographic and clinical data

Age, gender, disease onset, exercise intolerance, muscle contracture, myalgia, myoglobinuria, and weakness were analyzed and correlated to gene sequencing and mRNA expression. Disease severity was graded according to Martinuzzi's score as described before [6].

## Morphological features of muscle biopsies

Muscle samples from the left brachial biceps were stained with hematoxylin and eosin (H&E) in order to quantify the internal and normally positioned nuclei and number of vacuoles per sample; ATPase 9.4 was used to quantify fiber type predominance and measure muscle fiber diameter [5].

The set of histological and histochemical stains applied for all biopsies (suspect of McArdle and controls) was: H&E, Gomori, NADH, SDH, COX-SDH, PAS, ORO, phosphofructokinase. Also, the myophosphorylase stain, a qualitative histochemical test, was performed to obtain the diagnosis: the absence of staining was considered as negative for myophosphorylase reaction. Measurements were done according Dubowitz's method.

Approximately four to six fields (40x) were selected and analyzed clockwise for each patient's muscle sample. The same muscle biopsy was used to access morphological and *PYGM* expression data.

## Laboratory exams

We analyzed ammonia levels at rest and during grip strength, both obtained by MIFET. Grip strength was measured for each patient using an electronic hand dynamometer (CAMRY EH101). Additionally, at rest (just before the start of the ischemic test), we obtained the CK level as well as urea, creatinine and urine analyze.

## Gene sequencing

In order to identify *PYGM* (Gene ID: 608455) variants, DNA was extracted from blood or muscle samples with the QIAamp DNA Blood Kit/DNeasy blood and tissue kit according to the manufacturer's recommendations (Qiagen). Subsequently, the variant c.148C>T located in exon 1 and the variants c.1537A>G; c.1827G>A; c.2075_2076insAAA located at exons 13, 15, and 17, and previously described in Brazilian patients, were screened by Sanger sequencing, performed in an ABI3500 equipment (Life Technologies) according to the manufacturer's recommendations. The primers were generated according to Kubisch [10].

Patients that showed no variant by specific Sanger sequencing were selected for next generation sequencing (NGS) methodology using the TruSight Inherited Disease panel (Illumina, Inc) or ClearSeq Inherited diseases panel (Agilent Technologies, Inc). The experiments were performed according to the manufacturer's protocol. Sequence variants were evaluated using Varstation® tool (https://varstation.com/) considering the transcript NM_005609.1. The considered variants by NGS were the ones observed more than 10x in the sample, with frequency of 0.3 to 0.7 for heterozygosis.

After analysis of variants, samples were classified by the number of pathogenic alleles (zero, one, or two) causing premature termination codon (PTC).

## *PYGM* mRNA expression

Total RNA was isolated from muscle biopsies using TRIzol reagent (Thermo Fisher Scientific) and complementary DNA (cDNA) was synthetized from 0.5 μg total RNA, using the QuantiNova Reverse Transcription Kit (Qiagen), according to the manufacturer's recommendations.

The specific *PYGM* primers were designed using the Primer 3 Input 0.4.0 software, available at http://frodo.wi.mit.edu/primer3/. Their specificity was verified by Primer-BLAST [11,12]. The differences in primer sequence and stringency conditions of the reaction mean that the primer pair designed does not amplify *PYGL* or *PYGB*. The reference gene *RPL13α*

was used to normalize mRNA values. Primer sequences and the amplicon sizes are presented in S1 Table.

Quantitative real-time polymerase chain reaction (RT-qPCR) was performed with the 7500 Real-Time PCR System (Applied Biosystems, Foster City, CA, EUA). Temperature cycling included a hot start at 95ºC for 10 min followed by amplification: 40 cycles of 95ºC for 15 sec and 60ºC for 25 sec. The reactions were prepared in a final volume of 15 µL and contained: 1X SYBR Green mix (QuantiNOVA SYBR Green PCR kit, QIAGEN cat. no. 208056), 0.2 µM of specific primer, and 1.5 µL cDNA.

Relative *PYGM* expression for case and control samples was calculated with the $2^{-\Delta Ct}$ method [13]. mRNA expression was also compared to the number of PTC alleles observed in the sample. Patients' samples were classified according to the degree of PYGM expression, compared to the average expression of this gene in the control group: "hypoexpression" was considered when PYGM values of $2^{-DCt}$ were below the standard deviation in the control group. Measurements within the limits of the standard deviation values in the control group were considered "normoexpression", and values of $2^{-DCt}$ of PYGM above the standard deviation in the control group were considered "hyperexpression".

## Statistical analysis

Descriptive analysis was performed to determine central tendency and dispersion for the quantitative variables and absolute and relative frequencies for the qualitative variables. The Shapiro-Wilk test was used to analyze the quantitative normally distributed data. Student's *t* test was also used to analyze the association among *PYGM* mRNA expression and other clinical and laboratory aspects. The level of significance considered was 5%. Kruskall-Wallis test was used to compare *PYGM* expression to the number of PTCs in the sample. Dunn's nonparametric test was used to correct multiple analysis data.

Quantitative variables as the number of Type 1 and Type 2 fibers were presented by means and 95% confidence intervals, since the variables had a normal distribution (Shapiro-Wilk test, p> 0.05). To test the hypothesis that patients with McArdle have more type 2 fibers than type 1, Pearson's correlation test (r) was used.

Statistical analysis was performed with GraphPad Prism and Stata® Software (StataCorp, LC) version 11.0.

## Statement attesting consent

Written informed consent for DNA and RNA analysis was obtained from all the recruited patients when primary diagnostic procedures were performed, with explicit consent for future use for research purposes, according to the Declaration of Helsinki and the study was approved by our local Ethics Committee: CEPES Research protocol number 408.414 [14].

## Results

The mean age at diagnosis of the 15 McArdle's patients evaluated was 41.4 years (ranging from 20 to 60 years old). All patients refer that symptoms begin in the childhood, but couldn't precise the age of onset. The characteristics of the sample population as demographic features, main symptoms, laboratorial results and morphological findings are described at Table 1. The mean age of control group was 39.0 years.

When we compared Martinuzzi's score and mRNA expression we obtained an inverse relation between both parameters. The average of mRNA expression was 0.47, 0.30 and 0.13 for the scores 1, 2 and 3, respectively. However, no significant difference was observed among the samples, possible due to the low number of samples in each category (Table 2).

**Table 1. Laboratorial and morphological aspects of the Brazilian sample of McArdle's disease patients.**

| Data | n = 15 | % |
|---|---|---|
| Gender (male) | 7 | 58.3 |
| Fiber type predominance | | |
| Type 1 | 3 | 20.0 |
| Type 2 | 12 | 80.0 |
| Martinuzzi's score | | |
| 1 | 5 | 33.3 |
| 2 | 8 | 53.3 |
| 3 | 2 | 13.3 |
| Signs and symptoms % | | |
| Fatigue | 15 | 100 |
| Muscle contracture | 10 | 66.7 |
| Myalgia | 12 | 83.3 |
| Exercise intolerance | 15 | 100 |
| Dark urine | 10 | 66.7 |
| | Mean (SD) | Min—Max |
| Age (years) | 41.4 (10.4) | 20–60 |
| % fibers with vacuoles | 23.9 (15.8) | 1.99–58.3 |
| % internal nuclei | 9.4 (4.3) | 1.39–15.7 |
| Mean muscle fiber diameter Type 1 | 59.7 (13.5) | 32.6–84.2* |
| Mean muscle fiber diameter Type 2 | 60.8 (11.2) | 35.4–75.4* |
| Grip strength (kg) | 36.5 (24.6) | 9.9–86.9 |
| CK diagnosis (IU/L) | 16,914 (22,848) | 840–90000 |
| CK rest (IU/L) | 2860(2990) | 439–10000 |
| Ammonia T0 | 46.56 (33) | 18–101 |
| Ammonia T1—T0 | 168.72 (125) | 16.5–323 |

CK: creatine kinase; SD: standard deviation* according to literature[5].

About morphological findings, there was no significant difference between the percentage of Type 1 and type 2 fibers (r = -0.076). However, 80% (n = 12) of the McArdle's patients had more type 2 muscle fibers than type 1. The mean muscle fiber diameter exhibited a normal size variation. Considering sequencing findings, the main *PYGM* variant was c.148C>T (PTC), observed in eight/15 of the samples (53.3%). Other nine variants were also observed but at a low frequency, being present in one or two patients each one. For two patients, only variants with uncertain clinical significance were identified in the exons tracked, and for one patient, only a benign variant was found. Genomic position, predicted effect, and database annotations for the variants are available as supplementary data (S2 Table), Clinical and molecular findings per sample were observed in Table 2.

Regarding *PYGM* mRNA expression, eight patients (53.3%) presented decreased expression, while seven (46.6%) demonstrated normal expression, (considering the average value of control samples) as showed in Table 1. Considering gene variants, eight/15 patients had two PTC alleles; three/15 had one, and four/15 patients had unknown number of PTCs. All McArdle's patients showed 0.56-fold change in *PYGM* expression in relation to control group. Patients with unknown number of PTC presented 0.98-fold change, while patients with one and two PTCs showed 0.40 and 0.42, respectively, in relation to control group (Fig 1). However, the lowest level of mRNA expression was observed in a patient with one PTC and one synonymous variant (sample D/ Table 2).

**Table 2. Descriptive data between McArdle's patients and controls considering PTC, age, PYGM expression and Martinuzzi's score.**

| patient number | allele 1 | allele 2 | PTC | Age (years) | Martinuzzi's scale | % fibers with vacuoles | % internal nucleos | Mean muscle fiber diameter Type 1 | Mean muscle fiber diameter Type 2 | Fiber predominance | 2^-ΔCt PYGM | Control number | Age | 2^-ΔCt PYGM |
|---|---|---|---|---|---|---|---|---|---|---|---|---|---|---|
| 1 | c.1975C>A c.2024C>T | c.2123A>T | Unknown | 20 | 2 | 9,4 | 4,3 | 58,1 | 57 | 2 | 0,739524866 | 1 | 31 | 0,356975966 |
| 2 | c.148C>T | c.1827G>A | 1 | 36 | 2 | 2 | 6,8 | 53 | 46,5 | 2 | 0,014440541 | 2 | 40 | 0,490635322 |
| 3 | c.148C>T | c.148C>T | 2 | 60 | 2 | 14,5 | 10 | 84,2 | 62,3 | 1 | 0,632138307 | 3 | 19 | 0,443555514 |
| 4 | c.1975C>A | c.2123A>T | Unknown | 40 | 1 | 8,7 | 10 | 60,1 | 60 | 2 | 0,424075767 | 4 | 47 | 0,433387027 |
| 5 | c.1827+7A>G | unknown | Unknown | 41 | 1 | 28,6 | 1,4 | 58,3 | 63,9 | 2 | 0,431975814 | 5 | 51 | 0,507523106 |
| 6 | c.148C>T | c.148C>T | 2 | 38 | 2 | 32,5 | 8,2 | 63,6 | 63,3 | 2 | 0,03155577 | 6 | 17 | 0,547470697 |
| 7 | c.148C>T | c.2392T>C | 1 | 40 | 2 | 8,1 | 4,8 | 47,3 | 59,6 | 2 | 0,207584346 | 7 | 45 | 0,662868037 |
| 8 | c.148C>T | c.148C>T | 2 | 48 | 3 | 36,5 | 11,5 | 55,7 | 67,6 | 1 | 0,022955461 | 8 | 54 | 0,398614249 |
| 9* | c.148C>T | c.148C>T | 2 | 53 | 2 | 20,5 | 14,7 | 39,6 | 45,6 | 2 | 0,053613634 | 9 | 32 | 0,725042077 |
| 10 | c.1948C>T | c.613G>A | 1 | 51 | 1 | 28,5 | 5,3 | 64,8 | 61,1 | 1 | 0,314984149 | 10 | 6 | 0,489113993 |
| 11 | c.148C>T | c.148C>T | 2 | 25 | 2 | 15,2 | 11,5 | 32,6 | 34,6 | 2 | 0,033338528 | 11 | 45 | 0,251600846 |
| 12 | c.527A>C | unknown | Unknown | 36 | 2 | 58,3 | 13,9 | 78,4 | 78,2 | 2 | 0,16287767 | 12 | 46 | 0,353383757 |
| 13* | c.148C>T | c.148C>T | 2 | 47 | 2 | 45,9 | 14 | 61 | 56,6 | 2 | 0,310863673 | 13 | 65 | 0,231478539 |
| 14# | c.148C>T | c.148C>T | 2 | 48 | 2 | 34,8 | 15,7 | 72,4 | 75,4 | 2 | 0,2842699 | 14 | 58 | 0,522570582 |
| 15# | c.148C>T | c.148C>T | 2 | 38 | 1 | 15,3 | 9,5 | 65,9 | 72,1 | 2 | 0,150906275 | 15 | 16 | 0,632180931 |
| Total | | | | 40 | 75 | 23,4 | 10,3 | 59,6 | 60,2 | 80% type 2 | | | | 0,477816048& |

PTC: premature termination codon; PYGM: glycogen phosphorylase muscle* and #: identifications of individuals from the same family.

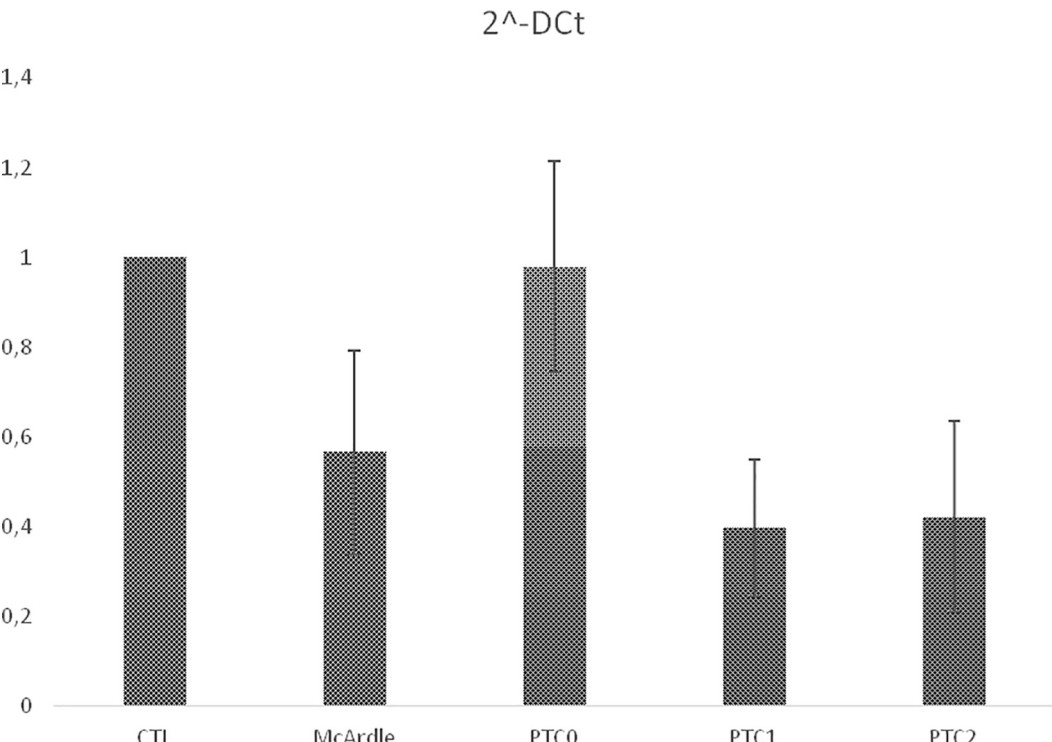

**Fig 1. *PYGM* differential expression among groups.** Legend: CTL: Control group; McArdle: All patients; PTC0: Patients with unknown number premature termination codon; PTC1: Patients with one premature termination codon; PCT2: Patients with two premature termination codon.

Among the samples there were two families with two affected patients each one (brothers and sisters) and two PTCs. In one family, we observed that clinical severity and gene expression varied between brothers (Table 2).When analyzing the association between *PYGM* mRNA expression and other laboratory and morphological aspects, there was no statistically significant association (S3 Table; p > 0.05).

## Discussion

This study compared different aspects of McArdle's disease: clinical, demographic, laboratorial and genetic data in relation to *PYGM* mRNA expression. We also included a quantitative morphological analysis from analysis of muscle biopsies. *PYGM* mRNA expression was not related to clinical or laboratory data. Thus, we believe that a different mechanism may be responsible for different variables observed among the patients.

Concerning morphological analysis, it was observed a predominance of type 2 muscle fibers (rich in glycogen) in McArdle's patients. Our hypothesis is that it could occurs due to compensatory effect in response to the injured fibers. To date, very few reports exist about fiber type composition muscle fiber type of in McArdle patients, specifically. Felice et al [15] reported type 1 muscle fiber atrophy in three patients with McArdle's disease. In contrast, Kohn et al [16], in an elegant study, analysed the three MHC isoforms (I, IIa and IIx) in the McArdle samples and compared them with the control ones. They did not find difference in isoform expression between the McArdle patients and healthy controls. Hereafter, new studies should investigate muscle fibre composition of McArdle patients.

Despite *PYGM* expression was not homogeneous in our sample, the ROC curve analysis showed high specificity to detect McArdle's cases, suggesting that *PYGM* hypoexpression could represent a biomarker to detect the disease. However, the validation of this biomarker may require an increased number of patients to be analyzed.

Since McArdle disease is thought to be an autosomal recessive disorder, all patients should have a pathogenic variant in both copies of the *PYGM* gene (homozygous or compound heterozygous) [7]. Sequencing analysis allowed us to identify ten different *PYGM* coding sequence variants, half of them already described as pathogenic and already observed in Brazilian and European patients.

The most frequent variant found among the samples was c.148C>T, a result that corroborates Gurgel-Giannetti's findings in a previous study of Brazilian McArdle patients [8]. Besides, the world literature, describes this variant occurring over 50% of Caucasian patients [17,18]. This variant is associated with a mechanism known as "nonsense-mediated mRNA decay" (NMD), which promotes degradation of mRNA transcripts that contain PTCs [16], which would partly explain the absence of myophosphorylase activity observed in muscle biopsies.

Another pathogenic variant causing PTC, c.1948C>T, was found in one patient, following the same mechanism described above. Additionally, the synonymous variant observed in one sample was previously associated to the formation of abnormal mRNA splicing species [19]. The other ones were missense variants and the mechanism associated to pathogenicity depends on the variant position.

Considering the number of PTCs in each sample, we observed in most of the cases an inversely proportional relationship between PTCs number and *PYGM* mRNA expression. We also noticed that patients with missense variants had RNA expression similar to controls and a similar phenotype to patients with one PTC. There were some outliers in the sample that did not follow the aforementioned general trends.

Nogales-Gadea [18] perceived that some variants did not follow the NMD rules completely, noticing that some PTC variants did not decrease *PYGM* RNA expression and some missense variants resulted in RNA decay. They made a correlation between cDNA appearance and the distance of PTC to the end of the transcript and suggested that missense variants could produce a mark in the transcript which could be detected by the mechanism of mRNA degradation.

García-Consuegra [20], evaluated the unexpected consequences of missense mutations in McArdles muscles. They observed that 95% of patients, irrespective of the *PYGM* genotype, had no glycogen phosphorylase in their muscles and proposed that missense mutations could likely alter PYGM mRNA secondary structure, leading to ribosome stalling and subsequent degradation. Our findings corroborate their observations.

Three patients in our sample showed no pathogenic variant in *PYGM* however, the clinical history, muscle biopsy findings and MIFET were compatible with McArdle's diagnosis. For one patient, an intronic variant or CNV could be responsible for phenotype. For the other two, VUS can be responsible for pathogenicity. We cannot to discard the existence of tissue specific gene variants, not observed in this study.

Here, NGS was effective as a diagnostic tool in 80% of our patients so, clinical phenotype associated with muscle biopsy is still a useful tool for McArdle disease diagnosis when the molecular test does not detect pathogenic variants.

Overall, our findings revealed that mRNA expression alone is not useful as a predictive factor responsible for prognosis of this disorder. The severity of McArdle disease may be associated with different mechanisms, including post-transcriptional events, epigenetics factors, or protein function.

## Supporting information

**S1 Table. Primer sequences and expected amplicons for *PYGM* and *RPL13α (internal control).*** Legend: bp: base pair
(DOCX)

**S2 Table. Brazilian McArdles patients' *PYGM* variations.**
(DOCX)

**S3 Table. Association between *PYGM* mRNA expression and different laboratorial and morphological variables.**
(DOCX)

## Author Contributions

**Conceptualization:** Alzira A. S. Carvalho, Matheus M. Perez, Itatiana Rodart, David Feder.

**Data curation:** Alzira A. S. Carvalho, Denise M. Christofolini, David Feder.

**Formal analysis:** Alzira A. S. Carvalho, Denise M. Christofolini, Matheus M. Perez, Beatriz C. A. Alves, Karine C. Turke, David Feder, Fernando L. A. Fonseca.

**Methodology:** Alzira A. S. Carvalho.

**Project administration:** Alzira A. S. Carvalho, David Feder, Fernando L. A. Fonseca.

**Software:** Matheus M. Perez, Itatiana Rodart, Francisco W. S. Figueiredo, Karine C. Turke.

**Validation:** Alzira A. S. Carvalho, Denise M. Christofolini, Matheus M. Perez, Beatriz C. A. Alves, Itatiana Rodart, Francisco W. S. Figueiredo, Karine C. Turke, David Feder, Marcondes C. F. Junior, Ana M. Nucci, Fernando L. A. Fonseca.

**Visualization:** Alzira A. S. Carvalho, Denise M. Christofolini, Matheus M. Perez, Beatriz C. A. Alves, Itatiana Rodart, Francisco W. S. Figueiredo, Karine C. Turke, David Feder, Marcondes C. F. Junior, Ana M. Nucci, Fernando L. A. Fonseca.

**Writing – original draft:** Alzira A. S. Carvalho, Denise M. Christofolini.

**Writing – review & editing:** Denise M. Christofolini, David Feder, Fernando L. A. Fonseca.

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
