## [Decision Letter · Decision Letter 0]

29 Apr 2020

PONE-D-20-09063

PYGM mRNA expression in McArdle disease: demographic, clinical, morphological and genetic features

PLOS ONE

Dear Dr Alves de Siqueira Carvalho,

Thank you for submitting your manuscript to PLOS ONE. After careful consideration, we feel that it has merit but does not fully meet PLOS ONE’s publication criteria as it currently stands. Therefore, we invite you to submit a revised version of the manuscript that addresses the points raised during the review process.

We would appreciate receiving your revised manuscript by Jun 13 2020 11:59PM. To enhance the reproducibility of your results, we recommend that if applicable you deposit your laboratory protocols in protocols.io, where a protocol can be assigned its own identifier (DOI) such that it can be cited independently in the future. For instructions see: http://journals.plos.org/plosone/s/submission-guidelines#loc-laboratory-protocols

We look forward to receiving your revised manuscript.

Kind regards,

Gisela Nogales-Gadea, Ph.D.

Academic Editor

PLOS ONE

Journal Requirements:

"NO - The funders had no role in study design, data collection and analysis, decision to

publish, or preparation of the manuscript."

Additional Editor Comments (if provided):

Dear Authors,

It has been a great opportunity to be an academic editor of your manuscript. I appreciate the effort that you have done in trying to bring more light to the McArdle disease field. However, in my opinion and in the opinion of the reviewers, your manuscript needs some major modifications:

There is one major absent in your article, that is to have considered this publication: Missense mutations have unexpected consequences: The McArdle disease paradigm. Hum Mutat. 2018. In this manuscript it is reported that the lack of phenotype-genotype relationship, may be explained by a mechanism controlling translation of messengers containing PYGM mutations, which was blocking protein production in mostly all patients. So this would be the explanation of the absence of genotype-phenotype correlation. In relation with this publication, did you have chance to analyse muscle glycogen protein by western blot in the muscle biopsies of your patients? These results would be a high contribution, given our current knowledge of the disease.

Other aspect to improve in your manuscript, is that the information it is presented in a very disorganized and diverse manner. For example, you talk about normoexpression and hipoexpression in Table 2, but there is no mention of the criteria for this classification in the manuscript. In supplementary Table 3 and 4, patients are identified with letters and in the next table with numbers.

Other aspects:

- This affirmation in the abstract it is not true: “mRNA expression and its association with clinical, morphological, and genetic aspects of the disease have not been studied previously”. The mRNA expression and its association with genetic aspects of the disease have been studied previously as you mention in your paper.

-Table 2: this table for me does not add any important information to the manuscript, I would just remove it.

-Supplementary Table 3 and 4: There is information repetitive between these two tables, that should be fused in one. And this information should become not supplementary but a Table of your manuscript.

-There is no sense in making a ROC curve for differentiating patients from controls, when you have so many patients which PYGM expression levels similar to the controls (I will just eliminate this form the article).

-In the patients in which you have not found all the PYGM mutations (there are deletions or intronic mutations no possible to detect with the techniques you have been used), you can not say that they are normal, you can just mention mutation not found or PYGM variation identified with unknown effect. Also, in these patients, I can not understand how you can assess the PTC number, if you have no idea of the mutations that these patients are carrying or you have just a mutation of unknown effect, you can not have the information on the PTC number.

- It would be very important to add to your tables the age at onset on the disease. An enormous delay in the diagnosis of this patients have been described. I would add this information in the fuse supplementary table 3 and 4.

- In your histological analysis you mention a fiber predominance of type 2 among the patients. However, there is not raw data on the muscle biopsy/per patient, however you can find it of fiber diameter, although no differences have been found at this level. Can you supply these information so readers can understand what is the magnitude of this difference. Also, in the discussion you should add and mention your results in relation with this article: “McArdle disease does not affect skeletal muscle fibre type profiles in humans”. Biol Open. 2014

Reviewers' comments:

Reviewer's Responses to Questions

**Comments to the Author**

1. Is the manuscript technically sound, and do the data support the conclusions?

Reviewer #1: Partly

Reviewer #2: Partly

2. Has the statistical analysis been performed appropriately and rigorously? 

Reviewer #1: Yes

Reviewer #2: I Don't Know

3. Have the authors made all data underlying the findings in their manuscript fully available?

Reviewer #1: Yes

Reviewer #2: Yes

4. Is the manuscript presented in an intelligible fashion and written in standard English?

Reviewer #1: Yes

Reviewer #2: No

5. Review Comments to the Author

Reviewer #1: The article is well written and technically sound, and the clinical and molecular data recorded from patients is truly very interesting. However, what it fails to me is the hypothesis of the manuscript. Why try to link Pygm mRNA levels to clinical phenotype without any consideration to protein levels? Most Pygm mutations cause a complete absence of myophosphorylase protein, and thus causing the disease phenotype. Besides, in the manuscript is described as an inclusion criteria "the lack of myophosphorylase staining in the muscle biopsies", this is, no myophosphorylase protein. Then, if all 15 patients had a complete absence of myophosphorylase protein, why different Pygm mRNA levels should be related to a different clinical phenotype? Or at least, why didn't you analyse it by western blot? This is for me a major issue that should be answered and clarified. If the authors feel this reviewer has not completely understood their point, please discuss it.

Besides this major point, there are minor issues that should be taken also in consideration:

-Abstract line 26: PYGM (glycogen phosphorylase gene)-muscle should be added (Muscle glycogen phosphorylase gene)

-Demographic and Clinical Data line 86: There is no necessity to describe in detail all the Martinuzzi scale grades, with the mention of reference (6) is more than enough.

-Pygm mRNA expression line 124: Even though it is specified that brachial biceps was used for morphology analysis, it is not specified whether the same muscle was used for mRNA analysis. Although one can imagine that the same biopsy was used for both things, it must be specified.

Reviewer #2: Dear Authors,

It is very interesting to read a Brazilian study in a rare disease. I would suggest a few adjustments to improve the quality of the manuscript as listed below:

Line 45: suggestion: to replace the term “cramps” by “muscle contracture”

Line 47: please correct grammar “ at rest”

Line 55: there has been research corelating sedentariness and McArdle symptoms severity – if the authors would like to quickly comment it in the intro to be more accurate

Line 63: prospective study based on case notes (for the clinical phenotype)? Why this study was considered a prospective one? Please clarify

Line 70: did the NMD symptoms of the control group overlap with McArdle disease phenotype?

Line 73 - 78: Please review English grammar of this whole paragraph – example: “Diagnosis inclusion criteria of McArdle disease”

Line 79: ICF for what? Genetic testing, or case report publication, or was this part of a research project? Perhaps it worth linking this section with line 154 (section 1.9) and keep the ICF info altogether

Please review the concept of muscle cramps x muscle contracture in McArdle

Line 96: I suppose the most important aspects of the muscle biopsy is to describe how the absence myophosphorylase was assessed, as it was one of the criteria for the diagnosis – or used to exclude a McArdle disease diagnosis in the control group. Did you assess regenerating fibers?

Line 102: what’s the grip strength assessment? Did you mean you assessed ammonia as part of the forearm exercise test?

Line 201: “None of our data was  related to PYGM expression.” If those patients have mcardle disease and the phenotype matches the genetic diagnosis, why would you state that your data is not related to PYGM? Please clarify what you mean and re-write this statement.

Line 205: please provide reference for this hypothesis

Line 239: phosphorylase stain may fade. Please confirm why these 3 patients had mcardle and not a faded staining – did you stain this biopsy with a healthy control at the same slide to assess technical issues and misdiagnosis?

Did you assess lactate? Why it was not described in table 1?

Why you did not use the hallmark of McArdle disease do confirm the phenotype? (the second wind phenomenon)

Why the lack of lactate following exercise was not reported or used as a diagnostics criteria?

In general I believe you may have a very interesting manuscript to be published, but it would be great if you could please review key concepts of GSDV, muscle biopsy staining/GSDV diagnosis and the manuscript writing.

6. PLOS authors have the option to publish the peer review history of their article (what does this mean?). If published, this will include your full peer review and any attached files.

Reviewer #1: No

Reviewer #2: No

---

## [Author Response · Author response to Decision Letter 0]

14 May 2020

Response to Reviewers: uploaded

---

## [Decision Letter · Decision Letter 1]

16 Jun 2020

PONE-D-20-09063R1

PYGM mRNA expression in McArdle disease: demographic, clinical, morphological and genetic features

PLOS ONE

Dear Dr. Alves de Siqueira Carvalho,

Thank you for submitting your manuscript to PLOS ONE. After careful consideration, we feel that it has merit but does not fully meet PLOS ONE’s publication criteria as it currently stands. Therefore, we invite you to submit a revised version of the manuscript that addresses the points raised during the review process.

We look forward to receiving your revised manuscript.

Kind regards,

Gisela Nogales-Gadea, Ph.D.

Academic Editor

PLOS ONE

Additional Editor Comments (if provided):

Thanks a lot for addressing all the reviewers and editors comments. You have done a great work in your manuscript, and I think your paper has been improved considerably.

Regarding my part, I will just suggest not to include this "there is a moderate correlation (r between 0.5 and 0.7) but with a

significant r for p <0.10 and not for p <0.05.Below, you see the raw data with 15 patients: I could include them in supplementary data"

Reviewer 1 has some more comments, so please address them.

Reviewers' comments:

Reviewer's Responses to Questions

**Comments to the Author**

1. If the authors have adequately addressed your comments raised in a previous round of review and you feel that this manuscript is now acceptable for publication, you may indicate that here to bypass the “Comments to the Author” section, enter your conflict of interest statement in the “Confidential to Editor” section, and submit your "Accept" recommendation.

Reviewer #1: (No Response)

2. Is the manuscript technically sound, and do the data support the conclusions?

Reviewer #1: No

3. Has the statistical analysis been performed appropriately and rigorously? 

Reviewer #1: Yes

4. Have the authors made all data underlying the findings in their manuscript fully available?

Reviewer #1: Yes

5. Is the manuscript presented in an intelligible fashion and written in standard English?

Reviewer #1: Yes

6. Review Comments to the Author

Reviewer #1: As far as we know what causes McArdle disease is the absence of myophosphorylase protein. Thus, I still don't see why different mRNA levels should lead to different disease severity if the consequence, regardless of the different mRNA levels, is the same, this is, no protein. So, why do you think that different mRNA levels should modify the phenotype? If you think that besides its coding role (nonsense here because it does not lead to any functional protein) the remaining 40-50% of mRNA may have a non-coding function modifying the disease phenotype, you should clear mention it as an hypothesis. But I don't share your rationale that is better to analyse the mRNA levels because is more "sensitive" than western blot, as I don't see what is the sensitivity for if the detectable mRNA levels do not serve to any purpose in terms of protein production?

7. PLOS authors have the option to publish the peer review history of their article (what does this mean?). If published, this will include your full peer review and any attached files.

Reviewer #1: No

---

## [Author Response · Author response to Decision Letter 1]

19 Jun 2020

RESPONSE TO REVIEWER 1

Reviewer #1: As far as we know what causes McArdle disease is the absence of myophosphorylase protein. Thus, I still don't see why different mRNA levels should lead to different disease severity if the consequence, regardless of the different mRNA levels, is the same, this is, no protein. So, why do you think that different mRNA levels should modify the phenotype? If you think that besides its coding role (nonsense here because it does not lead to any functional protein) the remaining 40-50% of mRNA may have a non-coding function modifying the disease phenotype, you should clear mention it as an hypothesis. But I don't share your rationale that is better to analyse the mRNA levels because is more "sensitive" than western blot, as I don't see what is the sensitivity for if the detectable mRNA levels do not serve to any purpose in terms of protein production?

Answer: Yes. McArdle's disease is confirmed by the absence of myophosphorylase in the muscle after histochemical analysis of the biopsy. However, what explains the different presentations of the condition among patients? The aim of our study was to verify whether the amount of mRNA could be correlated with clinical characteristics or with gene variants. Since mRNA is the protein's precursor, the rational of our study was that gene expression analysis would be more sensitive than histochemical analysis to detect small amounts of gene product, which could be responsible for small protein production, explaining thus the differences between patients. However, our hypothesis has not been confirmed. We found no correlation between mRNA levels and symptoms but we found a partial correlation with genetic findings. The mRNA levels, although lower in patients than in controls, were not directly related to the severity of the disease.

---

## [Editor Report · Decision Letter 2]

10 Jul 2020

PYGM mRNA expression in McArdle disease: demographic, clinical, morphological and genetic features

PONE-D-20-09063R2

Dear Dr. Alves de Siqueira Carvalho,

We’re pleased to inform you that your manuscript has been judged scientifically suitable for publication and will be formally accepted for publication once it meets all outstanding technical requirements.

Kind regards,

Gisela Nogales-Gadea, Ph.D.

Academic Editor

PLOS ONE
---

## [Editor Report · Acceptance letter]

17 Jul 2020

PONE-D-20-09063R2 

PYGM mRNA expression in McArdle disease: demographic, clinical, morphological and genetic features 

Dear Dr. Alves de Siqueira Carvalho:

I'm pleased to inform you that your manuscript has been deemed suitable for publication in PLOS ONE. Congratulations! Your manuscript is now with our production department. 

Kind regards, 

on behalf of

Dr. Gisela Nogales-Gadea 

Academic Editor

PLOS ONE